# Anti-Methanogenic Effect of Phytochemicals on Methyl-Coenzyme M Reductase—Potential: In Silico and Molecular Docking Studies for Environmental Protection

**DOI:** 10.3390/mi12111425

**Published:** 2021-11-19

**Authors:** Yuvaraj Dinakarkumar, Jothi Ramalingam Rajabathar, Selvaraj Arokiyaraj, Iyyappan Jeyaraj, Sai Ramesh Anjaneyulu, Shadakshari Sandeep, Chimatahalli Shanthakumar Karthik, Jimmy Nelson Appaturi, Lee D. Wilson

**Affiliations:** 1Vel Tech High Tech Dr. Rangarajan Dr. Sakunthala Engineering College, Anna University, Chennai 600062, India; jiyyappan.biotek@gmail.com (I.J.); drsairamesh@gmail.com (S.R.A.); 2Department of Chemistry, College of Sciences, King Saud University, P.O. Box 2455, Riyadh 11451, Saudi Arabia; 3Department of Food Science and Biotechnology, Sejong University, Seoul 05006, Korea; arokiyaraj16@sejong.ac.kr; 4Department of Chemistry, S J College of Engineering, JSS Science and Technology University, Mysuru 570006, India; sandeep12chem@gmail.com (S.S.); csk@jssstuniv.in (C.S.K.); 5School of Chemical Sciences, Universiti Sains Malaysia, Gelugor 11800, Malaysia; jimmynelson@usm.my; 6Department of Chemistry, University of Saskatchewan, Saskatoon, SK S7N 5C5, Canada

**Keywords:** ruminants, methane emission, methyl-coenzyme M reductase, phytochemicals, in-silico screening, molecular docking

## Abstract

Methane is a greenhouse gas which poses a great threat to life on earth as its emissions directly contribute to global warming and methane has a 28-fold higher warming potential over that of carbon dioxide. Ruminants have been identified as a major source of methane emission as a result of methanogenesis by their respective gut microbiomes. Various plants produce highly bioactive compounds which can be investigated to find a potential inhibitor of methyl-coenzyme M reductase (the target protein for methanogenesis). To speed up the process and to limit the use of laboratory resources, the present study uses an in-silico molecular docking approach to explore the anti-methanogenic properties of phytochemicals from *Cymbopogon* *citratus*, *Origanum vulgare*, *Lavandula officinalis*, *Cinnamomum zeylanicum*, *Piper betle*, *Cuminum cyminum*, *Ocimum gratissimum*, *Salvia sclarea*, *Allium sativum*, *Rosmarinus officinalis* and *Thymus vulgaris*. A total of 168 compounds from 11 plants were virtually screened. Finally, 25 scrutinized compounds were evaluated against methyl-coenzyme M reductase (MCR) protein using the AutoDock 4.0 program. In conclusion, the study identified 21 out of 25 compounds against inhibition of the MCR protein. Particularly, five compounds: rosmarinic acid (−10.71 kcal/mol), biotin (−9.38 kcal/mol), α-cadinol (−8.16 kcal/mol), (3*R*,3a*S*,6*R*,6a*R*)-3-(2*H*-1,3-benzodioxol-4-yl)-6-(2*H*-1,3-benzodioxol-5-yl)-hexahydrofuro[3,4-*c*]furan-1-one (−12.21 kcal/mol), and 2,4,7,9-tetramethyl-5decyn4,7diol (−9.02 kcal/mol) showed higher binding energy towards the MCR protein. In turn, these compounds have potential utility as rumen methanogenic inhibitors in the proposed methane inhibitor program. Ultimately, molecular dynamics simulations of rosmarinic acid and (3*R*,3a*S*,6*R*,6a*R*)-3-(2*H*-1,3-benzodioxol-4-yl)-6-(2*H*-1,3-benzodioxol-5-yl)-hexahydrofuro[3,4-*c*]furan-1-one yielded the best possible interaction and stability with the active site of 5A8K protein for 20 ns.

## 1. Introduction

Global warming has become an increasing concern to sustaining life on our planet. Various factors have led to global warming, especially the emission of greenhouse gases from various biotic and abiotic sources. One of the most concerning greenhouse gases is methane (CH_4_) which has a 28-fold greater warming potential over carbon dioxide (CO_2_) [1]. Though anthropogenic emission of greenhouse gases of 7% to 18% is generally accepted, the increasing trend of emissions has raised global concern [2,3]. When investigating the sources of methane emissions, a large contribution of emissions trace back to livestock, and especially to ruminants such as sheep and cattle. The production of methane by ruminants is referred to as ruminant methanogenesis. This methanogenesis is also a factor for loss of energy in ruminants which could have been otherwise used for growth development or that of milk production [4]. The process of methanogenesis in the ruminants is outlined as the conversion of CO_2_ and H_2_ into CH_4_ which is carried out by the ruminant microbiome which includes archaea, bacteria, fungi, and protozoa [5,6]. Studies on the ruminant microbiome reveal that bacteria inhabit 95% of the microbiome and 2–4% of archaea and 1% of protozoa and fungi [7]. For methanogenesis, rumen archaea bacteria require the enzyme methyl coenzyme M reductase (MCR). Therefore, the MCR protein was used as a marker of methanogenesis in diverse environments [8,9,10]. The most common methanogens (hydrogenotropicarchae) are from the genus Methanobrevibacter, closely related to methane emissions [11,12]. Some of the microorganisms involved in ruminant methane production are *Methanobacterium bryantii* [8], *Methanomicrobium mobile* [9], *Methanobrevibacter olleyae, Methanobrevibacter millerae* [10], *Methanobacterium formicicum* [11], *Methanobrevibacter ruminantium* [12], and *Methanobrevibacter gottschalkii* [4]. These microorganisms possess MCR, which is involved in the methanogenesis metabolism and referred to as a hydrogenotrophic process [13]. MCR is a critical enzyme involved in the final step of nearly all methanogenesis-based metabolism, along with coenzyme M (CoM) which is 2-merceptoethane sulfonic acid. MCR aids in the reduction of methylated CoM, which results in the production of methane. This shows that methyl-CoM reductase is an important enzyme which plays a significant role in methanogenesis [14]. This has led researchers to find a significant target molecule for inhibiting this enzyme so as to reduce methane production in ruminants. A significant source of compounds for various applications have been provided by nature’s best progeny: plants and their related phytochemicals. Globally, various plants have been studied and documented for various biological activity and other applications. Selected phytochemicals from various plants can be studied for their inhibitory activity against this critical enzyme as a solution for attenuating ruminant methane production. However, every plant possesses numerous phytochemicals, each with their respective functions and methane emission. To address the increasing global emissions in an effective manner, there is a need for promising and rapid testing methods to save time and resources for in-vitro or laboratory screening of these phytoactive compounds against MCR. Thus, in-silico studies can afford screening of numerous phytochemicals against this enzyme so as to find a potential inhibitor of MCR without the requirement of in-vitro laboratory resources. Molecular docking is one of the most used in-silico screening activities of any biomolecules or ligands with the target protein. Thus, the current study involves the process of selecting highly bioactive phytochemicals based on previous literature and screening of compounds for potential inhibitory activity against methyl-CoM reductase. The target phytochemicals with promising results can be used for in-vitro studies that can be further used as phytogenic feed additives for ruminants to ensure reduced methane emissions.

## 2. Materials and Methods

### 2.1. Plant Selection

An extensive literature study was carried out to identify suitable plants with highly bioactive phytochemicals that can be utilized as a potential inhibitor for rumen methane production. Initially, phytochemical databases such as Dr. Duke’s Phytochemical and Ethnobotanical Dictionary of Natural Products 27.2 [15], IMPPAT: Indian Medicinal Plants, Phytochemistry And Therapeutics [16], Super Natural Database V2 [17] were taken into consideration and based on the collected literature data, 11 indigenous plants were chosen, as follows: *Cymbopogon citratus*, *Origanum vulgare*, *Lavandula officinalis*, *Cinnamomum zeylanicum*, *Piper betle*, *Cuminum cyminum*, *Ocimum gratissimum*, *Salvia sclarea*, *Allium sativum*, *Rosmarinus officinalis* and *Thymus vulgaris*. These plants were selected for rumen methanogenesis analysis.

### 2.2. Primary Database Preparation

The selected plants were again subjected to literature analysis for enumeration of phytochemicals present, mainly through GCMS reports published in reputed peer review journals [18,19,20,21,22,23,24,25,26,27,28]. A primary database of bioactive phytochemicals was prepared using Molecular operating Environment software (MOE version–2015.10–Chemical Computing Group: Montreal, QC, Canada).

### 2.3. Virtual Screening of Ligands

A virtual screening of ligands was carried out to scrutinize the database for the identification of compounds with higher bioactivity and low toxicity. The ADME properties of the compounds were predicted using the Swiss ADME web server and Lipinski rule of five along with “drug likeliness” predictions that were used to filter out the best compounds with required activity [29].

### 2.4. Molecular Docking

Phytochemical compounds which showed nil violation for Lipinski rule of five and moderate “drug likeliness” were selected for the process of molecular docking. Docking was carried out to evaluate the real time interaction between the phytochemical compounds with the target protein of interest (methyl-coenzyme M reductase, catalyst for rumen methane production) by measuring the binding energy of the complex (drug with target protein). AutoDock 4.0 (AD) and AutoDock Vina 1.5.6 (ADV) (Centre for Computational Structural Biology, La Jolla, CA, USA) were used for this process and the results were visualized using the Chimera UCSF visualization tool (Resource for Biocomputing, Visualization, and Informatics (RBVI), San Francisco, CA, USA).

#### 2.4.1. Ligand Preparation and Target Protein

The three-dimensional structure of the target protein (methyl-coenzyme M reductase) was retrieved from the protein data bank using its PDB ID and it was visualized using discovery studio visualizer (DSV) software. Water molecules, ions, and standard inhibitors if present were also removed using the discovery studio visualization tool [30]. Marvin sketch software was used to construct the three-dimensional structures of the ligands to be studied. Avogadro (molecular modelling tool) was used for energy minimization of ligand molecules to obtain the best pose with low energy (Most stable 3D form). Both local charge and torsion (rotational motion) was provided to three-dimensional ligand structure for obtaining an effective binding complex with target protein. Similarly, Kollmann charge and polar hydrogen bonds were added to the protein [31]. Three dimensional PDBQT files of both the ligand and the protein were generated using MGL tools.

#### 2.4.2. Protocol of Docking Studies

*Auto dock*: Auto dock version 4.0 software was used for automated docking studies. AutoGrid grid, a component of auto dock, was used to compute the grid maps with the interaction energies depending upon the macromolecule target of the docking study. The grid center was placed on the active target site of the protein (predicted using prankweb.cz online tool) [32]. Then, the binding free energy of the inhibitors was evaluated using automated docking studies. The best conformations search was done by adopting a genetic algorithm with local search (GA-LS) method. The docking parameters were set at default values with 100 independent docking runs using the software ADT (AutoDock Tool Kit). Root mean square (RMS) tolerance of 2.0 Å was performed using structures generated after completion of docking via cluster analysis. Molecular graphics and visualization were performed with the UCSF Chimera package [33,34].

*Autodock vina*: By using an Autodock vina software, binding energies were reported and determined in kcal/mol unit for ligand–protein. The grid box was generated by targeting the active site with a size of (x = 40, y = 40, z = 40) and a center of (x = 54.17, y = 9.53, z = 37.09) was set to cover the binding site for 5A8K protein, which was adopted and the conformations simulated. Virtual molecular docking and analysis were performed using the AutoDock Vina 1.5.6 tool. All the ligand molecules were docked to the active site of 5A8K protein. Finally, protein–ligand interactions were analyzed by using Biovia discovery studio visualizer. The top-ranked ligands with the most negative, favorable interactions were carried forward for further analysis of molecular dynamics simulations.

### 2.5. Molecular Dynamics Simulation Studies

The molecular dynamic investigation was performed for the top two compounds (3*R*,3a*S*,6*R*,6a*R*)-3-(2*H*-1,3-benzodioxol-4-yl)-6-(2*H*-1,3-benzodioxol-5-yl)-hexahydrofuro-[3,4-*c*]furan-1-one and rosmarinic acid, which were obtained from the molecular docking. This approach was used to analyze the ligand consistency and the binding mode stability in the binding pocket of protein [35]. To maintain the system’s neutral conditions, the minimum quantity of Na^+^ ions with salt ions were added. Once the compound attained energy minimization, it was put through equilibrium using NPT gathering for 2 ns [36]. The relaxed system was then submitted to 20 ns MD simulations, which were performed using Marlyna-Tobias-Klein barostats at 1 bar pressure and a Nosé–Hoover thermostat set to 300 K under an NPT ensemble. In each course of the box, the protein complexes had a minimum of 7 Å buffer to allow for significant fluctuations along the MD simulation. By using desmond modules of the Schrodinger 2020-2 suite software (New York, NY, USA), the molecular dynamics simulation study was carried out [37].

## 3. Results and Discussion

### 3.1. Plant Selection and Database Preparation

Phytochemical compounds recognized from GCMS reports of existing literature and collected from various databases were used to prepare a primary database for the study. Initially, a total of 168 potential chemical entities (data not shown) were identified as bioactive phyto-ingredients present in 11 indigenous medicinal plants. Two dimensional structures of the compounds were downloaded from pubchem and were compiled into a single structural database as shown in the Appendix A.

### 3.2. Virtual Screening of Ligands

The virtual screening of ligands was carried out using the Swiss ADME server and parameters like Lipinski’s rule of five and “drug likeliness” were used as a filter for narrowing down the number of compounds to be taken up for docking. Initially, among 168 compounds present in the database, a total of 51 compounds with less than three violations were filtered out. Then these compounds were subjected to the same Lipinski filter (without any violation) along with “drug likeliness” and finally 25 compounds were found to have nil violations in Lipinski rule and also zero to one violation for “drug likeliness”, proving it as a lead target molecule for docking studies. The compounds selected through virtual screening are listed in Table 1.

### 3.3. Molecular Docking of Compounds

The three-dimensional structure of methyl-coenzyme M reductase (Figure 1) was retrieved from the protein data bank using the PDB ID 5A8K (https://www.rcsb.org/structure/5A8K, accessed on 17 October 2021), which was prepared for docking analysis using discovery studio visualizer software. Additional data is presented in the Appendix A related to the hydrogen bond interactions of ligands with Methyl co-enzyme M reductase. The protein consists of six chains (chains A–F), whereas chain A, B, C represent a dimer unit of chain D, E, F.

The 25 ligands taken for the study were also preprocessed by the addition of charge and torsion with the help of AutoDock tools 4.0. The active site of the protein was predicted using PrankWeb server and the selected residues were used as coordinates for generating the AutoGrid. The active site that was predicted by the PrankWeb server is tabulated in Table 2 and the active pocket is illustrated in Figure 1.

The docking results of potent ligands identified through virtual screening and docked against Methyl Coenzyme M are summarized in Table 3. The binding energy (kcal/mol) and inhibition constant (Ki) were used to evaluate the binding affinity of the inhibitors.

Analysis of the docking results of the phytochemical compounds revealed that most of the compounds possessed good binding affinity against Methyl co-enzyme M reductase, which proved better inhibition. Prominent results were observed for compounds such as Ferulic-acid, Niacin, *p*-Hydroxy-benzoic-acid, Vanillic-acid, Biotin, Caffeic-acid, Gallic-acid, Rosmarinic-acid, 1,3,8-*p*-Menthatriene, Methyl trans-geranylacetate, 2,4,7,9-Tetramethyl-5decyn-4,7-diol, 2,6-Bis(3,4methylenedioxyphenyl)-3,7-dioxabicyclo (3.3.0) octane, α-cadinol, epi-Cubebol, and Eugenol. As shown in Figure 2, (3*R*,3a*S*,6*R*,6a*R*)-3-(2*H*-1,3-benzodioxol-4-yl)-6-(2*H*-1,3-benzodioxol-5-yl)-hexahydrofuro[3,4-*c*]furan-1-one had the highest inhibition energy with −12.21 kcal/mol (AD) and 8.6 kcal/mol (ADV) against Methyl co-enzyme M reductase. Similarly, Rosmarinic-Acid (cf. Figure 3) and Biotin had a binding energy of −10.71 (AD) kcal/mol, −8.5 kcal/mol (ADV), −9.38 kcal/mol (AD), and −6.0 kcal/mol (ADV) respectively. Other notable ligands such as 2,4,7,9-tetramethyl-5-decyn4,7diol showed a binding energy of −9.02 kcal/mol (AD), −5.9 kcal/mol (ADV), and α-cadinol showed a binding energy of −8.16 kcal/mol (AD), −6.6 kcal/mol (ADV) against the target protein (cf. Figure 4).

Overall, the docking results revealed a very interesting trend that indicated that most of the phytochemicals chosen were shown to be good inhibitors for methane production. Out of 25 compounds studied for inhibition of methane production using docking, two compounds showed binding energy more than −10 kcal/mol, two compounds had binding energy around −9 kcal/mol, one compound had binding energy around −8 kcal/mol, and six compounds exhibited a binding energy of approximately −7 kcal/mol, while the remaining fourteen compounds had a binding energy of around −4.5 to −7 kcal/mol. Similarly, upon considering the number of hydrogen bonds formed, five compounds formed two hydrogen bonds with the target receptor, whereas sixteen compounds formed one hydrogen bond, and the remaining four compounds had no hydrogen bonding. If the effect of inhibition constant is taken into account, four compounds showed an inhibition constant at the nanomolar (nM) level, whereas twenty-one compounds had the inhibition constant at micromolar (μM), which revealed that the ligands under observation had very good binding with the target protein.

Upon considering the various parameters of docking such as binding energy, the number of hydrogen bonds, and inhibition constant, it was clearly evident that the selected phytochemicals have greater specificity towards the methyl-CoM reductase binding site and could serve as potent anti-methanogen inhibitors.

### 3.4. Molecular Dynamics Studies

Molecular dynamics simulation (MDS) provides a virtual approach for finding the behavior of each system in real time and understanding the conformational and most effective active analogues in the active site of the protein. The confirmation by MDS may also be used to determine which amino acid residues in the protein active site are involved in the interaction with a ligand. The interactions were computed as a time element using the data from the MDS, RMSD, and RMSF. According to the docking interaction, RMSD shows compound (3*R*,3a*S*,6*R*,6a*R*)-3-(2*H*-1,3-benzodioxol-4-yl)-6-(2*H*-1,3-benzodi-oxol-5-yl)-hexahydrofuro[3,4-*c*]furan-1-one with 5A8K had a good interaction and docking results with the methanothermobacter wolfeii (5A8K) protein. Compound (3*R*,3a*S*,6*R*,6a*R*)-3-(2*H*-1,3-benzodioxol-4-yl)-6-(2*H*-1,3-benzodioxol-5-yl)-hexahydrofuro[3,4-*c*]furan-1-one had a low interaction up to 2.5 ns (0.4 Å to 2.8 Å) and it showed a good interaction up to 20 ns (2.3 Å to 3.2 Å) with 5A8K, and while compound rosmarinic-acid showed a good interaction up to 2 ns (0.8 Å to 2.0 Å), it showed a very low interaction up to 20 ns (0.8 Å to 3.2 Å) with 5A8K, as shown in Figure 5. The RMSF graph of compound (3*R*,3a*S*,6*R*,6a*R*)-3-(2*H*-1,3-benzodioxol-4-yl)-6-(2*H*-1,3-benzodioxol-5-yl)-hexahydrofuro[3,4-*c*]furan-1-one showed that the significant fluctuations were observed initially, as follows: residues number 50 up to 4.5 Å, residue number 180 up to 4.5 Å, residue number 260 up to 4.3 Å, residue number 550 up to 4.5 Å, and residue number 940 up to 4.0 Å got the maximum deviation. While the RMSF graph of compound rosmarinic-acid showed that residue number 60 up to more than 4.5 Å, residue number 550 up to 4.5 Å, and residue number 610 up to 4.3 Å got the maximum deviation throughout the MD simulations. The remaining residues were known to be moderately stable and fluctuating well below 2.0 Å. Compound (3*R*,3a*S*,6*R*,6a*R*)-3-(2*H*-1,3-benzodioxol-4-yl)-6-(2*H*-1,3-benzodioxol-5-yl)-hexahydrofuro[3,4-*c*]furan-1-one had an H-bonding interaction of up to 0.025 with 5A8K of GLN183 and 0.01 with ARG383 and THR265 amino acids, while compound 9 had an H-bonding interaction of up to 1.0 with GLN395 and HIS156, 0.90 with GLY119, 0.60 with ARG120 and 0.25 with ARG401 and GLN332 amino acids. In the remaining areas, water bridge bonding and hydrophobic bonds were visible. Frequently, compound (3*R*,3a*S*,6*R*,6a*R*)-3-(2*H*-1,3-benzodioxol-4-yl)-6-(2*H*-1,3-benzodioxol-5-yl)-hexahydrofuro[3,4-*c*]furan-1-one had 34% interaction with the GLY187 and GLY189, respectively. Similarly, compound rosmarinic-acid had 98% interaction with GLN395, 77% with PHE396, 62% with ARG120 and 88% with GLY119 and HIS156 residues. However, both the results were found to be converging after 20 ns of simulated time and it can be concluded that compound (3*R*,3a*S*,6*R*,6a*R*)-3-(2*H*-1,3-benzodioxol-4-yl)-6-(2*H*-1,3-benzodioxol-5-yl)-hexahydrofuro[3,4-*c*]furan-1-one had greater interaction stability with active amino acids of 5A8K than the rosmarinic-acid. Figure 5 shows RMSD, RMSF, protein–ligand contacts and ligand–protein contacts of compound (3*R*,3a*S*,6*R*,6a*R*)-3-(2*H*-1,3-benzodioxol-4-yl)-6-(2*H*-1,3-benzodioxol-5-yl)-hexahydrofuro[3,4-*c*]furan-1-one and rosmarinic-acid with 5A8K protein, respectively.

## 4. Conclusions

With a steady increase in greenhouse gas production and severe natural calamities due to climatic change arising because of global warming, there is a need to find green solutions to address this issue, hence efforts were made to correlate the methanogenic activity of bioactive phytochemicals present in various indigenous medicinal plants. This study was carried out to determine the methanogenic activity of 11 medicinal plants. Using an extensive literature survey, a total of 168 bioactive phytochemicals were taken into consideration and virtual screening was carried out with ADME and Lipinski rules of five as filters. Finally, 25 compounds were screened, and molecular docking was carried out for the same against methyl co-enzyme M reductase as the target protein. Most of the compounds showed good results, where five compounds had exceptional binding affinity with low binding energy and better hydrogen bonding. The results support that phytochemicals such as (3*R*,3a*S*,6*R*,6a*R*)-3-(2*H*-1,3-benzodioxol-4-yl)-6-(2*H*-1,3-benzodioxol-5-yl)-hexahydrofuro[3,4-*c*]furan-1-one, rosmarinic-acid, biotin, 2,4,7,9-tetramethyl-5decyn-4,7-diol, and α-cadinol can be utilized for real time methanogenic applications. MD simulations were carried out for two compounds ((3*R*,3a*S*,6*R*,6a*R*)-3-(2*H*-1,3-benzodioxol-4-yl)-6-(2*H*-1,3-benzodioxol-5-yl)-hexahydrofuro[3,4-*c*]furan-1-one and rosmarinic-acid). Improved docking energies revealed the stability of protein–ligand complexes and the existence of hydrogen bonds. However, further in-vitro and in-vivo studies are encouraged to confirm the anti-methanogenic ability of the selected plant compounds.

## Figures and Tables

**Figure 1 micromachines-12-01425-f001:**
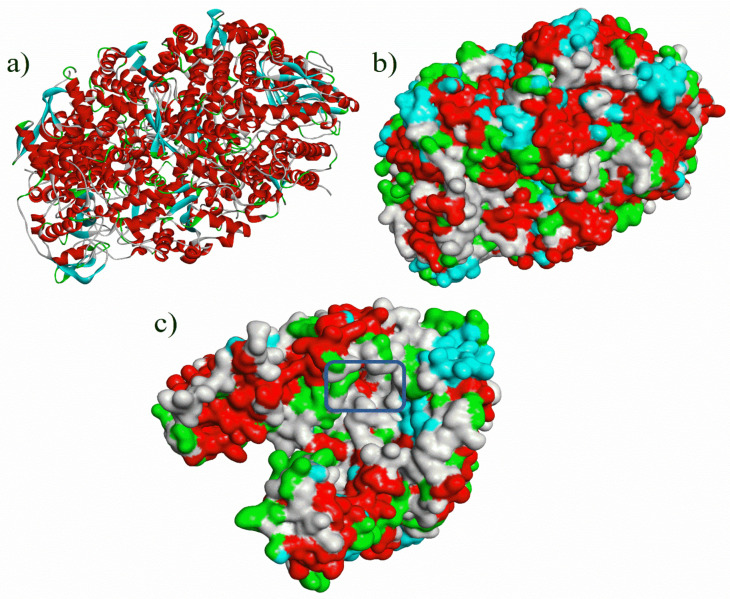
3D structure of Methyl co-enzyme M reductase (PDB ID—5A8K) (**a**) 3D protein ribbon structure of 5A8K (chains A to F); (**b**) 3D protein ribbon structure with surface pockets (chains A to F); and (**c**) 3D surface structure of active pocket of Methyl co-enzyme M reductase (PDB ID—5A8K) (chains A to C).

**Figure 2 micromachines-12-01425-f002:**
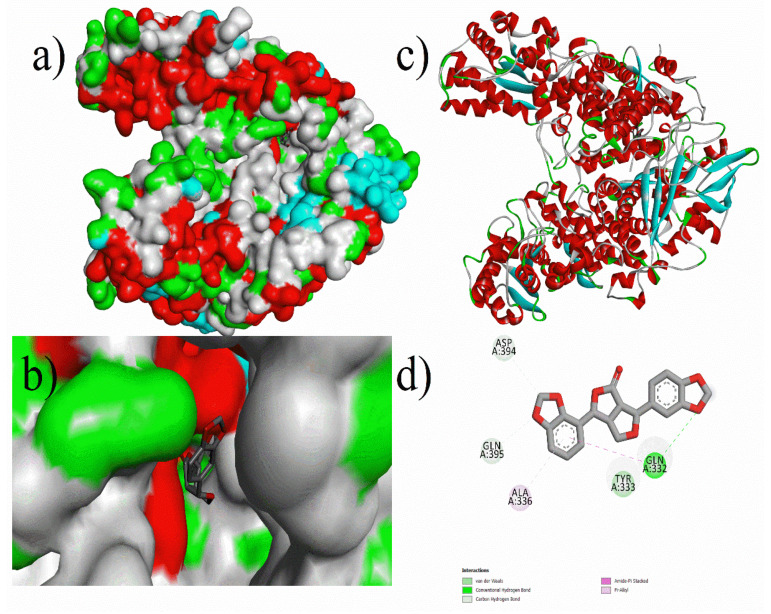
Molecular docking of (3*R*,3a*S*,6*R*,6a*R*)-3-(2*H*-1,3-benzodioxol-4-yl)-6-(2*H*-1,3-benzodioxol-5-yl)-hexahydrofuro[3,4-*c*]furan-1-one with 5A8K protein: (**a**) Docking surface pocket pose, (**b**) docking interaction at surface, (**c**) 3D protein-ligand interaction, and (**d**) 3D interaction of protein amino acid residues with the guest compound ((3*R*,3a*S*,6*R*,6a*R*)-3-(2*H*-1,3-benzodioxol-4-yl)-6-(2*H*-1,3-benzodioxol-5-yl)-hexahydrofuro[3,4-*c*]furan-1-one).

**Figure 3 micromachines-12-01425-f003:**
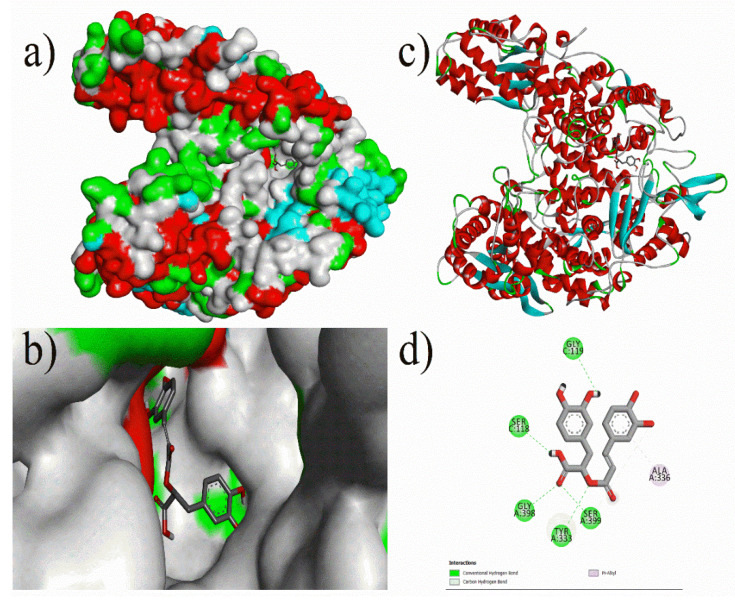
Molecular docking of Rosmarinic-acid with 5A8K protein: (**a**) Docking surface pocket pose, (**b**) docking interaction at surface, (**c**) 3D protein–ligand interactions, and (**d**) 3D interaction of protein amino acid residues with compound rosmarinic-acid.

**Figure 4 micromachines-12-01425-f004:**
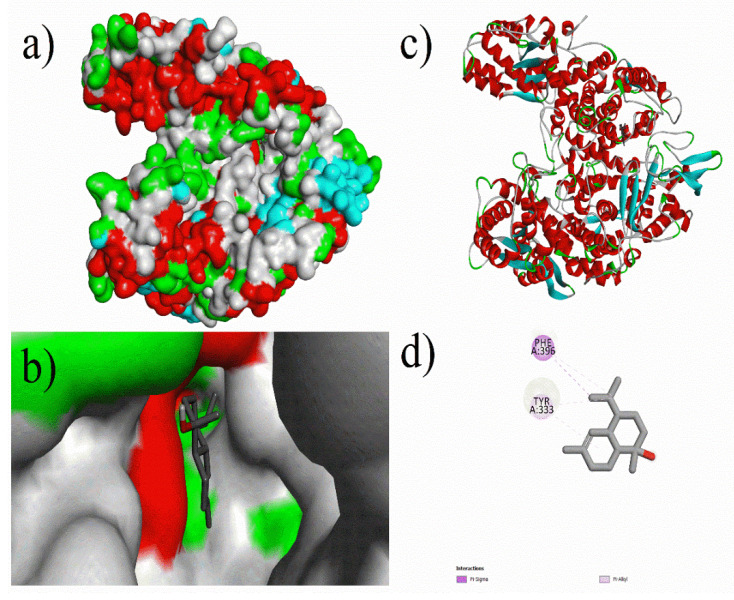
Molecular docking of α-cadinol with 5A8K protein: (**a**) Docking surface pocket pose, (**b**) docking interaction at surface, (**c**) 3D protein–ligand interaction, and (**d**) 3D interaction of protein amino acid residues with compound α-cadinol.

**Figure 5 micromachines-12-01425-f005:**
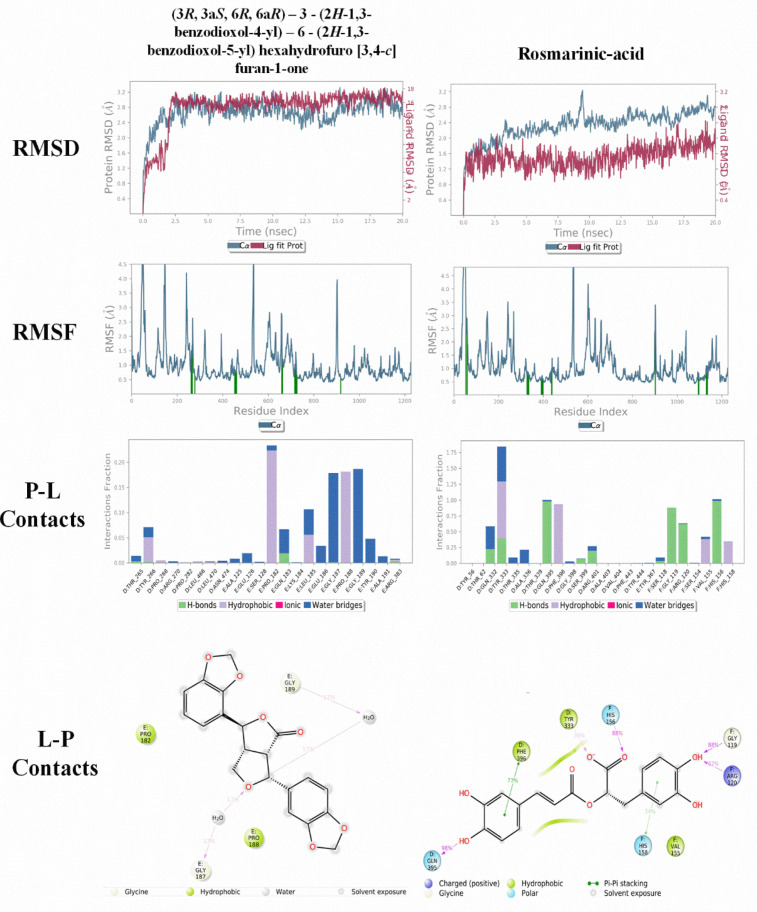
Illustration to show the outcomes of molecular dynamic simulation of compound (3*R*,3a*S*,6*R*,6a*R*)-3-(2*H*-1,3-benzodioxol-4-yl)-6-(2*H*-1,3-benzodioxol-5-yl)hexahydrofuro [3,4-*c*]furan-1-one and rosmarinic acid complexes. RMSD, RMSF of the backbone over the 20 ns MDS at 300 K of the complex systems. Protein (5A8K) ligand interaction (P-L contacts) and 2D-Diagram of Ligand Protein (L-P Contacts) interaction.

**Table 1 micromachines-12-01425-t001:** Compounds selected through virtual screening (Swiss ADME).

Molecule	Formula	MW	Lipinski Violations	Lead Likeness Violations
Cinnamic-acid	C_9_H_8_O_2_	148.16	0	1
Ferulic-acid	C_10_H_10_O_4_	194.18	0	1
Niacin	C_6_H_5_NO_2_	123.11	0	1
*p*-Hydroxy-benzoic acid	C_7_H_6_O_3_	138.12	0	1
Vanillic-acid	C_8_H_8_O_4_	168.15	0	1
Biotin	C_10_H_16_N_2_O_3_S	244.31	0	1
Caffeic-acid	C_9_H_8_O_4_	180.16	0	1
Gallic-acid	C_7_H_6_O_5_	170.12	0	1
Rosmarinic-acid	C_18_H_16_O_8_	360.31	0	1
Spathulenol	C_15_H_24_O	220.35	0	1
1,3,4-Eugenol	C_10_H_12_O_2_	164.2	0	1
1,3,8-p-Menthatriene	C_10_H_14_	134.22	0	1
1,3-Cyclopentadiene	C_5_H_6_	66.1	0	1
3,7-dimethyl-endo-borneol	C_10_H_18_O	154.25	0	1
Methyl trans-geranylacetate	C_13_H_20_O_3_	224.3	0	1
2,4,7,9-tetramethyl-5-decyn-4,7diol	C_14_H_26_O_2_	226.36	0	1
2,6-Bis(3,4methylenedioxyphenyl)-3,7-dioxabicyclo (3.3.0)octane	C_20_H_16_O_7_	368.34	0	1
2-Methyl-5-(1-propenyl)pyrazine	C_8_H_10_N_2_	134.18	0	1
α-cadinol	C_15_H_26_O	222.37	0	1
Diallyl tetrasulfide	C_6_H_10_S_4_	210.4	0	1
epi-Cubebol	C_15_H_24_O	220.35	0	1
Eugenol	C_10_H_12_O_2_	164.2	0	1
linalool	C_10_H_18_O	154.25	0	1
Pinacol	C_6_H_14_O_2_	118.17	0	1
Pulegone	C_10_H_16_O	152.23	0	1

**Table 2 micromachines-12-01425-t002:** Active site predicted for methyl-coenzyme M reductase (PDB ID—5A8K).

Chain	Residues Selected
Chain A	37 LYS, 62 THR, 64 LEU, 65 GLY, 67 ARG, 69 LEU, 70 MET, 72 TYR, 82 GLU, 83 GLY, 84 ASP, 87 HIS, 90 ASN, 267 LEU, 268 PRO, 269 VAL, 270 ARG, 272 ALA, 319 TRP, 320 LEU, 324 MET, 328 VAL, 329 GLY, 330 PHE, 331 THR, 332 GLN, 333 TYR, 336 ALA, 394 ASP, 395 GLN, 396 PHE, 397 GLY, 399 SER, 401 ARG, 403 ALA, 443 PHE, 444 TYR, 473 PRO, 474 ASN, 479 ALA, 480 MET, 481 ASN, 482 VAL
Chain B	49 ASN, 51 GLU, 52 GLY, 55 ASN, 111 ARG, 131 PRO, 169 GLU, 170 TYR, 173 ALA, 174 ASN, 175 ILE, 176 ALA, 177 THR, 178 MET, 179 LEU, 180 ASP, 181 ILE, 184 LYS, 186 GLU, 194 ASN, 196 MET, 199 HIS, 361 PHE, 362 PHE, 364 HIS, 365 SER, 366 ILE, 367 TYR, 368 GLY, 369 GLY, 374 ILE, 376 ASN, 378 ASN, 379 HIS, 380 ILE, 409 GLU, 410 ALA, 411 THR, 413 GLY, 414 LEU, 415 ILE, 417 GLU
Chain C	83 ARG, 84 TYR, 86 GLN, 117 LEU, 118 SER, 119 GLY, 120 ARG, 122 ILE, 124 GLU, 152 GLY, 153 LYS, 154 SER, 155 VAL, 156 HIS, 158 HIS, 171MET, 191 ILE

**Table 3 micromachines-12-01425-t003:** Docking results of the compounds against Methyl Coenzyme M.

Sample Number	Ligand	Binding Energy (kcal/mol)		Ki
		AD	ADV	
1	Cinnamic-acid	−4.44	−6.6	557.1 µM
2	Ferulic-acid	−6.70	−6.3	12.27 µM
3	Niacin	−6.19	−5.2	28.85 µM
4	*p*-Hydroxy-benzoic acid	−6.83	−5.9	9.82 µM
5	Vanillic-acid	−6.44	−6.0	18.91 µM
6	Biotin	−9.38	−6.0	0.132 µM
7	Caffeic-acid	−7.34	−6.3	4.17 µM
8	Gallic-acid	−7.83	−5.8	1.81 µM
9	Rosmarinic-acid	−10.71	−8.5	0.014 µM
10	1,3,8-*p*-Menthatriene	−6.90	−5.8	8.73 µM
11	1,3-Cyclopentadiene	−4.12	−4.4	954.6 µM
12	3,7-Dimethyl-endo-borneol	−6.81	−5.8	10.22 µM
13	Methyl trans-geranylacetate	−7.58	−7.4	2.78 µM
14	2,4,7,9-Tetramethyl-5decyn-4,7-diol	−9.02	−5.9	0.2465 µM
15	(3*R*,3a*S*,6*R*,6a*R*)-3-(2*H*-1,3-benzodioxol-4-yl)-6-(2*H*-1,3-benzodioxol-5-yl)-hexahydrofuro[3,4-*c*]furan-1-one	−12.21	−8.6	0.0012 µM
16	2-Methyl-5-(1-propenyl)pyrazine	−5.19	−5.4	158.1 µM
17	α-cadinol	−8.16	−6.6	1.04 µM
18	Diallyl tetrasulfide	−4.84	−3.7	283.6 µM
19	epi-Cubebol	−7.83	−6.3	1.82 µM
20	Eugenol	−7.08	−5.6	6.47 µM
21	Linalool	−6.80	−6.5	10.3 µM
22	Pulegone	−7.45	−6.1	3.49 µM
23	Pinacol	−6.33	−4.6	22.9 µM
24	Spathulenol	−4.45	−7.0	31.4 µM
25	2,6-Bis(3,4-methylenedi-oxyphenyl)-3,7-dioxabi-cyclo(3.3.0)octane	−5.18	−8.4	158.9 µM

AutoDock (AD) AutoDock Vina (ADV).

## Data Availability

The data presented in this study are available on request from the co-corresponding coauthor (R.J.). The data are not publicly available due to the raw/processed data required to reproduce these findings that cannot be shared at this time as the data also forms part of an ongoing study.

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
