# Peer review of "Anti-Methanogenic Effect of Phytochemicals on Methyl-Coenzyme M Reductase—Potential: In Silico and Molecular Docking Studies for Environmental Protection"

_micromachines, 2021, doi:10.3390/mi12111425_

Round 1

Reviewer 1 Report

Authors performed virtual screening and molecular docking simulations on methyl-coenzyme M reductase with phytochemicals from various plants. The goal of the study was to find novel candidates for the inhibition of MCR in ruminants as they are one of the main producers of the methane. While the study offers novel approach to solving the problem of the greenhouse gas emissions and thus reducing the global warming, there is still a lot of work to be done.

While the main concept of the study is scientifically sound, there were several flaws in the execution of the simulations that need to be adressed. Also the interpretation of results could be improved as there are several unclarities.

Major comments:

1) Authors state that they used 3D structure of the methyl-coenzyme M reductase with PDB ID 58AK. After quick check on the RCSB database, there is no entry with this ID. However, there is PDB ID 5A8K which is indeed structure of the methyl-coenzyme M reductase from Methanothermobacter wolfeii. However, in the Introduction section authors list 7 other microorganism involved in the ruminant methane production. It should be explained why was study performed on the crystral structure of another microorganism and what is structure similarity between MCR of Metanothermobacter wolfeii and MCR of microorganisms listed in the introduction.

2) In the methods section it is written that the crystal structure of the protein was used for docking and the preparation consisted out of adding charges and polar hydrogen bonds. I believe this is typo as one can only add polar hydrogen atoms to the crystal structure of the protein. Authors should describe here also how they treated protonation states of histidine residues (at least those in the active site). Moreover, considering only one structure (specially if it is structure taken directly from the PDB database) can give false results. I would recommend performing short classical MD simulation of the apo protein (~100 ns of production run)  to relax it and then cluster the trajectory in order to obtain more representative structure of the protein and perform the docking simulations on that/those structure. This process can also serve as a validation of the already obtained results.

3) When describing protocol of docking studies authors failed to mention what was the grid spacing and what was the overall size of the grid box. Moreover, the "active target site" is not described adequately. 

4) Since the docking was performed only on one strucutre/conformation of the protein, I would recommend performing short classical MD simulation (~100-200 ns of production run) on the ligand-protein complex and then perform analysis on the interactions between ligand and the protein.

5) Why was AutoDock version 4.0 used and not the newer version 4.2? Moreover, authors should cite AutoDock for using it. 

6) Authors explain results only in term of hydrogen bonds. Some compounds such as 4, 9, and 22 form two hydrogen bonds, yet their binding energy is less favourable than those of some compounds that form only one hydrogen bond. Additionally, some compounds do not form any hydrogen bond (i.e. compound 10) and they still have more favourable binding energy than some of the compounds that form one hydrogen bond. Authors should comment and explain those observations.

Moreover, interaction analysis between ligand and protein should also take into account hydrophobic interactions and pi-pi interactions. Finally, what were the hydrogen bond length and angle cut-offs in the hydrogen bond analysis?

7) It would interesting to know if there are already some known inhibitors for MCR and include those inhibitors in the docking calculations. Results (binding energies) from those calculations could serve as validation of the method as well as a good comparison for the binding energies of the phytochemicals studied.

Minor comments:

8) PDB ID 58AK should be corrected throughout the manuscript with the correct one and resolution of the crystal structure should be written. Moreover, correct PDB ID should be written already in the section 2.4.1 together with the citation.

9) Throughout the manuscript, consistent notation for PDB should be used (sometimes it is written PDB and sometimes pdb).

10) Line 48: ruminants, which is reffered --> ruminants is reffered

11) Lines 89-90: Authors should write the date when databases used in this study were accessed and/or downloaded.

12) Line 129: Auto dock --> AutoDock

13) Line 129: Auto grid --> AutoGrid

14) Line 130: auto dock --> AutoDock

15) Line 136: Auto-Dock Tool Kit --> AutoDockTools

16) General remark: Capitalisation of subsections should be consistent and not as it is now (sometimes only first charachter of the first word is written with capital letter, sometimes first characters of each word are written with capital letters, and in some cases the whole title is written with capital letters).

17) I would suggest moving Figure 1 to Supplementary.

18) Line 147: "(data not shown)" I believe this data is shown in the SI and this should be stated.

19) Table 1: I would suggest removing Canonical SMILES structure and adding the skeletal formulas of compounds. In Formula section, numbers should be written in subscript.

20) Line 169: AutoDock tool 4.0 --> AutoDockTools

21) Lines 170-171: What is generated is grid and not auto grid. But the software for generating the grid is AutoGrid.

22) Figure 2: Authors should mark where in the protein is located the active site and name chains A, B, and C on the figure.

23) Figure 3: I would suggest showing the active site in balls & sticks or lines representation together with the new cartoon representation for the backbone instead of the surface. Also the important residues should be named in the figure. Alternatively, figure with surface representation should be better explained.

24) Table 3: Bond length is written with too high precission. I would suggest using maximum two decimal numbers (this should also be applied to the main text and figure captions). Binding energies  should be always reported with same number of decimal numbers for all compound (same is true for Ki).

25) Figures 4-6: I would recommend to show ligand in one colour and protein in different one. Additionally, only important residues of the active site should be shown on each figure and they should be named. Finally, it would help the reader if all three figures are shown from the same perspective. Finally, I would suggest to make 2D diagrams of protein-ligand interactions, which would make results much more clear to the reader. 

26) Lines 181-183: This sentence is not clear to me. It is true that the results from docking simulations show good binding affinity, however it is difficult to say that this "proved better inhibition" as experiments have not been performed.

27) Lines 207-210: Binding energies should be written as negative numbers (-10 kcal/mol, -9 kcal/mol etc.)

28) Line 219: "...selected phytochemicals were shown to be an excellent inhibitor for rumen methane production". I cannot agree with this sentence as it is not supported with the experimental results. Also, simulations were performed on a single protein conformation without additional MD simulations.

Author Response

Authors Response to Reviewer Comments

Manuscript ID Micromachines-1387081

Reviewer #1

Authors performed virtual screening and molecular docking simulations on methyl-coenzyme M reductase with phytochemicals from various plants. The goal of the study was to find novel candidates for the inhibition of MCR in ruminants as they are one of the main producers of the methane. While the study offers novel approach to solving the problem of the greenhouse gas emissions and thus reducing the global warming, there is still a lot of work to be done.

While the main concept of the study is scientifically sound, there were several flaws in the execution of the simulations that need to be addressed. Also the interpretation of results could be improved as there are several unclarities.

Major comments:

Comment #1

Authors state that they used 3D structure of the methyl-coenzyme M reductase with PDB ID 58AK. After quick check on the RCSB database, there is no entry with this ID. However, there is PDB ID 5A8K which is indeed structure of the methyl-coenzyme M reductase from Methanothermobacterwolfeii. However, in the Introduction section authors list 7 other microorganism involved in the ruminant methane production. It should be explained why was study performed on the crystal structure of another microorganism and what is structure similarity between MCR of Metanothermobacterwolfeii and MCR of microorganisms listed in the introduction.

Response #1

We thank the reviewer for pointing the topological error. As per the reviewer, comment PDB ID changed to PDB ID 5A8K.

Metanothermobacterwolfeii was also added in the introduction and required citation for including the organism in the study and also taking the same for docking process was added.
The PDB file containing the structure was elucidated using experimental X-ray diffraction study rather than predicted modeling and also had a better resolution of 1.4 Å. So the PDB file 5A8K was selected for docking.

Comment #2

In the methods section it is written that the crystal structure of the protein was used for docking and the preparation consisted out of adding charges and polar hydrogen bonds. I believe this is typo as one can only add polar hydrogen atoms to the crystal structure of the protein. Authors should describe here also how they treated protonation states of histidine residues (at least those in the active site). Moreover, considering only one structure (especially if it is structure taken directly from the PDB database) can give false results. I would recommend performing short classical MD simulation of the apo protein (~100 ns of production run)  to relax it and then cluster the trajectory in order to obtain more representative structure of the protein and perform the docking simulations on that/those structure. This process can also serve as a validation of the already obtained results.

Response #2

Initially the faults in the residues were cleansed using the protein preparation wizard present in the Discovery studio.The above sentence is now included in the manuscript (Section 2.4.1). We agree with reviewer comment associated with MD simulation. We will take it into account in our future work. Thank you very much for your critical comment.

Comment #3

When describing protocol of docking study’s authors failed to mention what was the grid spacing and what was the overall size of the grid box. Moreover, the "active target site" is not described adequately. 

Response #3

            The grid size and the spacing of the grid box as now been mentioned in the result section of the manuscript along with detailed description.

Comment #4

Since the docking was performed only on one structure/conformation of the protein, I would recommend performing short classical MD simulation (~100-200 ns of production run) on the ligand-protein complex and then perform analysis on the interactions between ligand and the protein.

Response #4

            The molecular docking analysis was re-preformed to validate docking results and to reduce the conformational errors in the structure of the protein.We have limited computing resources to conduct the MD simulations. Your point on MD simulation is interesting, and we will consider it in our future work.

Comment #5

Why was AutoDock version 4.0 used and not the newer version 4.2? Moreover, authors should cite AutoDock for using it. 

Response #5

We used AutoDock version 4.0 for docking studies as per our earlier publication and a citation is now included as per your suggestion. (Section 2.4.2)

Ref. 35. Arokiyaraj, S., Bharanidharan, R., Agastian, P. et al. Chemical composition, antioxidant activity and antibacterial mechanism of action from Marsilea minuta leaf hexane: methanol extract. Chemistry Central Journal 12, 105 (2018). https://doi.org/10.1186/s13065-018-0476-4.

Thank you very much for your comment.

Comment #6

Authors explain results only in term of hydrogen bonds. Some compounds such as 4, 9, and 22 form two hydrogen bonds, yet their binding energy is less favourable than those of some compounds that form only one hydrogen bond. Additionally, some compounds do not form any hydrogen bond (i.e. compound 10) and they still have more favourable binding energy than some of the compounds that form one hydrogen bond. Authors should comment and explain those observations.

Response #6

Thank you point view based on the intermolecular interaction and binding energy and amino acid regions.

Comment #7

Moreover, interaction analysis between ligand and protein should also take into account hydrophobic interactions and pi-pi interactions. Finally, what were the hydrogen bond length and angle cut-offs in the hydrogen bond analysis?

Response #7

            A cut off angle of 20° and bond length saturation of 3 Å was used for generation of hydrogen bond by Chimera protein visualizer.

Comment #8

It would interesting to know if there are already some known inhibitors for MCR and include those inhibitors in the docking calculations. Results (binding energies) from those calculations could serve as validation of the method as well as a good comparison for the binding energies of the phytochemicals studied.

Response #8

Use of molecular docking approach in phytocompounds screening against rumen methane inhibition is new. Only limited study has been carried using docking approach (Arokiyaraj et al. 2020). We agree with reviewer comment, and in future, we use known methanogenic inhibitors for docking calculations.

Arokiyaraj S, Stalin A, Shin H. Anti-methanogenic effect of rhubarb (Rheum spp.) - An in silico docking studies on methyl-coenzyme M reductase (MCR). Saudi J Biol Sci. 2019 Nov; 26(7):1458-1462. doi: 10.1016/j.sjbs.2019.06.008. Epub 2019 Jun 13. PMID: 31762609; PMCID: PMC6864367.

Minor comments:

Comment #9

PDB .ID 58AK should be corrected throughout the manuscript with the correct one and resolution of the crystal structure should be written. Moreover, correct PDB ID should be written already in the section 2.4.1 together with the citation.

Response #9

Thank you point view suggestion carried out As per reviewer comment correction done.

Comment #10

Throughout the manuscript, consistent notation for PDB should be used (sometimes it is written PDB and sometimes pdb).

Response #10

The corrections were carried out throughout the manuscript as per the reviewer comment

Comment #11

Line 48: ruminants, which is reffered --> ruminants is reffered

Response #11

Correction made according to reviewer comments

Comment #12

Lines 89-90: Authors should write the date when databases used in this study were accessed and/or downloaded.

Response #12

Rewritten

Comment and Response #13:  Each of the line editing comments were addressed below

Line 129: Auto dock -->AutoDock - Done

Line 129: Auto grid -->AutoGrid - Done

Line 130: auto dock -->AutoDock - Done

Line 136: Auto-Dock Tool Kit -->AutoDockTools - Done

Comment #14

General remark: Capitalisation of subsections should be consistent and not as it is now (sometimes only first charachter of the first word is written with capital letter, sometimes first characters of each word are written with capital letters, and in some cases the whole title is written with capital letters). 

Response #14

Done

Comment #15

I would suggest moving Figure 1 to Supplementary.

Response #15

Figure 1 moved to supplementary section as Figure S1. Thank you very much for your suggestion.

Comment #16

Line 147: "(data not shown)" I believe this data is shown in the SI and this should be stated.

Response #16

Noted and Changed.

Comment #17

Table 1: I would suggest removing Canonical SMILES structure and adding the skeletal formulas of compounds. In Formula section, numbers should be written in subscript.

Response #17

Noted and Changed.

Comment and Response #18: Each of the line editing comments were addressed below

Line 169: AutoDock tool 4.0 -->AutoDockTools  -Done

Lines 170-171: What is generated is grid and not auto grid. But the software for generating the grid is AutoGrid.  - Done

Comment #19

Figure 2: Authors should mark where in the protein is located the active site and name chains A, B, and C on the figure.

Response #19

Done

Comment #20

Figure 3: I would suggest showing the active site in balls & sticks or lines representation together with the new cartoon representation for the backbone instead of the surface. Also the important residues should be named in the figure. Alternatively, figure with surface representation should be better explained.

Response #20

Done

Comment #21

Table 3: Bond length is written with too high precission. I would suggest using maximum two decimal numbers (this should also be applied to the main text and figure captions). Binding energies should be always reported with same number of decimal numbers for all compound (same is true for Ki).

Response #21

Table 3 was edited to address the reviewer comment.

Comment #22

Figures 4-6: I would recommend to show ligand in one colour and protein in different one. Additionally, only important residues of the active site should be shown on each figure and they should be named. Finally, it would help the reader if all three figures are shown from the same perspective. Finally, I would suggest to make 2D diagrams of protein-ligand interactions, which would make results much more clear to the reader.

Response #22

Done

Comment #23

Lines 181-183: This sentence is not clear to me. It is true that the results from docking simulations show good binding affinity, however it is difficult to say that this "proved better inhibition" as experiments have not been performed.

Response #23

The sentence was paraphrased according to the requirement

Comment #24

Lines 207-210: Binding energies should be written as negative numbers (-10 kcal/mol, -9 kcal/mol etc.)

Response #24

Done

Comment #25

Line 219: "...selected phytochemicals were shown to be an excellent inhibitor for rumen methane production". I cannot agree with this sentence as it is not supported with the experimental results. Also, simulations were performed on a single protein conformation without additional MD simulations.

Response #24

The sentence was paraphrased according to the requirement

In summary, we appreciate the insightful and constructive comments provided by Reviewer #1. As well, the manuscript was further edited for language, syntax, and clarity throughout.

Reviewer 2 Report

The manuscript by Dinakarkumar et al. describes molecular docking studies on phytochemicals from various 25 plants.

The manuscript aims to identify a good inhibitor for a specific protein target among molecules known in the literature. Unfortunately, however actual, the work is limited to a simple docking exercise that betrays the expectations.

Without going into the merits of the procedure adopted, the proposed work is too minimal, it lacks consistent results, and the analyses carried out do not achieve the goal set.

For this reason, I regret to inform the authors that the manuscript as presented is not worthy of publication.

To analyze the interactions established between the ligands and the receptor in detail, the authors should carry out molecular dynamics analyses. In addition, it would be useful also to calculate the residence time.

Reporting hydrogen bonds in a docking pose is inadequate to discriminate between binders and select the best inhibitor.

If the authors were to extend their investigations in the future, I would recommend fixing a few points in their manuscript.

  • The ligands they used were constructed from a smile format sequence. The author should specify whether they checked the chiral centers of the molecule (where present).
  • The authors used a rigid-rigid docking protocol. In the future, they might consider the flexibility of the binding site (also by simply including the flexibility of the side chains).
  • In an article, figures should help the reader understand several steps of the work and the caption of a figure should describe the image. Unfortunately, figures 2 and 3 seem not to meet these requirements.

For example, in figure 2, the authors wrote the following caption:

Figure 2. 3D structure of Methyl co-enzyme M reductase (pdb id - 58AK).

However, in figure 2, there are two images. What did the authors want to show? What should the reader understand?

The same problem with figure 3, the reader looking at a colored surface of the active site, what should he understand? Has it been mapped by charge or by what other property?

Ultimately, it would be appropriate to replace the ki given in the manuscript with kd. Unfortunately, docking algorithms can only give an estimate of the free energies of binding and dissociation constants.

It must be taken into account that the correlation Ki = dissociation constant of the enzyme-inhibitor complex = Kd given in the AutoDock 4.0 manual is only a rough approximation, and the prediction error is very high.

Author Response

Authors Response to Reviewer Comments

Manuscript ID Micromachines-1387081

Reviewer #2

The manuscript by Dinakarkumar et al. describes molecular docking studies on phytochemicals from various 25 plants.

The manuscript aims to identify a good inhibitor for a specific protein target among molecules known in the literature. Unfortunately, however actual, the work is limited to a simple docking exercise that betrays the expectations.

Without going into the merits of the procedure adopted, the proposed work is too minimal, it lacks consistent results, and the analyses carried out do not achieve the goal set.

For this reason, I regret to inform the authors that the manuscript as presented is not worthy of publication.

Comment #1

To analyze the interactions established between the ligands and the receptor in detail, the authors should carry out molecular dynamics analyses. In addition, it would be useful also to calculate the residence time.

Response #1

The molecular docking analysis was re-preformed to validate docking results and to reduce the conformational errors in the structure of the protein. We have limited computing resources to conduct the MD simulations. Your point on MD simulation is interesting, and we will consider it in our future work.

Comment #2

Reporting hydrogen bonds in a docking pose is inadequate to discriminate between binders and select the best inhibitor.

Response #2

According to the literature study, hydrogen bonding between proteins and ligands reveal the ligands' inhibitory properties. Thank you very much for your comment.

Comment #3

The ligands they used were constructed from a smile format sequence. The author should specify whether they checked the chiral centers of the molecule (where present).

Response #3

Done

Comment #4

The authors used a rigid-rigid docking protocol. In the future, they might consider the flexibility of the binding site (also by simply including the flexibility of the side chains).

Response #4

Corrections made in the manuscript according to reviewer comments

Comment #5

In an article, figures should help the reader understand several steps of the work and the caption of a figure should describe the image. Unfortunately, figures 2 and 3 seem not to meet these requirements.

Response #5

Rewritten accordingly

Comment #6

For example, in figure 2, the authors wrote the following caption:

Figure 2. 3D structure of Methyl co-enzyme M reductase (pdb id - 58AK).

Response #6:  The reviewers carried out the required correction

Comment #7

However, in figure 2, there are two images. What did the authors want to show? What should the reader understand?

The same problem with figure 3, the reader looking at a colored surface of the active site, what should he understand? Has it been mapped by charge or by what other property?

Response #7: Edits were carried out to address the reviewer comment for Fig. 3.

Comment #8

Ultimately, it would be appropriate to replace the ki given in the manuscript with kd. Unfortunately, docking algorithms can only give an estimate of the free energies of binding and dissociation constants.

It must be taken into account that the correlation Ki = dissociation constant of the enzyme-inhibitor complex = Kd given in the AutoDock 4.0 manual is only a rough approximation, and the prediction error is very high.

Response #8

We appreciate the reviewer suggestion on this point.

In summary, we appreciate the insightful and constructive comments provided by Reviewer #2. As well, the manuscript was further edited for language, syntax, and clarity throughout.

Reviewer 3 Report

The Authors have approach to the conducted the molecular docking studies of various plants which product highly bioactive compounds which can be investigated to find a potential inhibitor of methyl-coenzyme reductase. The manuscript deserves to publish in Micromachines after a minor correction. I would like to suggest introducing changes before publishing in Micromachines.

The authors should revise in the manuscript as the following points:

  1. Abstract: The abstract should state briefly the purpose of the research, the principle results and major conclusions. The abstract should be corrected.
  2. There is no clearly formulated goal and scope of work. Please correct it.
  3. Please complete conclusions to combine theoretical work with future or proven experimental work.

Author Response

Authors Response to Reviewer Comments

Manuscript ID Micromachines-1387081

Reviewer #3 

            The Authors have approach to the conducted the molecular docking studies of various plants which product highly bioactive compounds which can be investigated to find a potential inhibitor of methyl-coenzyme reductase. The manuscript deserves to publish in Micromachines after a minor correction. I would like to suggest introducing changes before publishing in Micromachines.

The authors should revise in the manuscript as the following points:

Comments

  1. Abstract: The abstract should state briefly the purpose of the research, the principle results and major conclusions. The abstract should be corrected.
  2. There is no clearly formulated goal and scope of work. Please correct it.
  3. Please complete conclusions to combine theoretical work with future or proven experimental work.

Response 

Corrections made in the manuscript according to reviewer comments

In summary, we appreciate the insightful and constructive comments provided by Reviewer #3. As well, the manuscript was further edited for language, syntax, and clarity throughout.

Round 2

Reviewer 1 Report

Authors improved the manuscript by addressing several minor and majority of the minor comments, however the major issues raised still remain and should be addressed adequately. Bellow is the list of them:

Comment #1: After reading the manuscript, I was not able to find any mention of the Metanobacterwolfeii in the Introduction section nor anywhere else in the manuscript despite the fact authors stated they added this part to the Introduction section. The connection between MCR of Metanobacterwolfeii (the crystal structure of the MCR) and 7 microorganisms involved in the methane production is still not explained. Moreover, I was also not able to find the reference for the used crystal structure in the manuscript.

Comments #2 & #4: I appreciate the fact that authors will consider MD simulations in their future work. However, my comments refer to the current manuscript and the MD simulations of the complexes obtained from docking simulations should be performed (alternatively, in case computational resources are limited authors could perform at least 1 MD simulation of the apo protein and then perform flexible docking simulations).

Comment #4: I could not find the results of the re-performed docking simulations. Maybe a Table in SI could be added where results of the first and second run of docking simulations are shown. Ideally, the second run of docking simulations should be performed with another docking software or at least with different scoring function.

Comment #6: This comment has not been addressed in the manuscript. It is still not clear to the reader why some compounds such as 4, 9, and 22 form two hydrogen bonds, yet their binding energy is less favourable than those of some compounds that form only one hydrogen bond. Additionally, some compounds do not form any hydrogen bond (i.e. compound 10) and they still have more favourable binding energy than some of the compounds that form one hydrogen bond. Authors should comment and explain those observations.

Comment #7: Analysis of hydrophobic and pi-pi interactions is still missing. It can be performed on the already obtained complexes.

Comment #12: Authors indeed added date, when the database was accessed (17/10/2021). However, this date is not correct one. The date when the database was accessed should be the one when it was accessed for the purpose of the study and not just when the manuscript was corrected.

Comment #20: I believe the Figure 2 still does not tell the reader anything about the active site of the enzyme.

Comment #21: Binding energies and Ki are still written with different number of decimal numbers. Also hydrogen bond lengths in the main text are still written with too high precision.

Comment #22: Figures 3-5 were not adapted in a clear enough way (interacting residues are still not named in the Figure and are shown only as ribbons, which does not explain the type of interaction to the reader).

Author Response

Author Response to Reviewer comments on MS ID: micromachines-1387081

Reviewer #1 (round 2)

Authors improved the manuscript by addressing several minor and majority of the minor comments; however, the major issues raised still remain and should be addressed adequately. Bellow is the list of them:

Comment #1: After reading the manuscript, I was not able to find any mention of the Metanobacterwolfeii in the Introduction section nor anywhere else in the manuscript despite the fact authors stated they added this part to the Introduction section. The connection between MCR of Metanobacterwolfeii (the crystal structure of the MCR) and microorganisms involved in the methane production is still not explained. Moreover, I was also not able to find the reference for the used crystal structure in the manuscript.

Response: For methanogenesis, rumen archaea bacteria require the enzyme methyl coenzyme M reductase (MCR). Therefore, MCR protein was used as a marker of methanogenesis in diverse environments Ref. 9-12 included in the revised MS. The most common methanogens (hydrogenotropicarchae) are from the genus Methanobrevibacter, closely related to methane emissions (Danielsson et al., 2012; Shi et al., 2014) [11, 12]. Evert C. Duin et al. (2016), reported that 3-nitrooxypropanol reduces the methane emissions from rumen methanogens in vitro (Methanobrevibacter sp., Methanobacterium bryantii, Methanothermobacter wolfeii and Methanosphaera) [13]. In this study we used MCR protein obtained from PDB 5A8K for in silico molecular docking study, also mentioned in supplementary file (cf. Page 3).

Comments #2 & #4: I appreciate the fact that authors will consider MD simulations in their future work. However, my comments refer to the current manuscript and the MD simulations of the complexes obtained from docking simulations should be performed (alternatively, in case computational resources are limited authors could perform at least 1 MD simulation of the apo protein and then perform flexible docking simulations).

Response: MD simulation analysis have been performed for complex (3R,3aS,6R,6aR)-3-(2H-1,3-benzodioxol-4-yl)-6-(2H-1,3-benzodioxol-5-yl)-hexahydrofuro[3,4-c]furan-1-one and rosmarinic acid. Both the complex molecules shows high favorable binding energy against 5A8K in docking analysis.

Comment #4: I could not find the results of the re-performed docking simulations. Maybe a Table in SI could be added where results of the first and second run of docking simulations are shown. Ideally, the second run of docking simulations should be performed with another docking software or at least with different scoring function.

Response: Three-dimensional structure of methyl-coenzyme M reductase (Figure 1) was retrieved from the protein data bank using the PDB id 5A8K and was prepared with the protein preparation tool of Discovery studios. 14 residues of chain A, 13 residues of chain B and 4 residues of chain C were shown to have multiple conformers. Therefore they were cleansed using the protein cleaning wizard of Discovery studio suite. Then the cleansed protein structure was subjected to MD simulation for 100 ns. The following protocol was followed for step-wise simulation to remove the structural constraints present in the protein molecule; Energy Minimization 1 to 20 ns, Energy Minimization 2 to 20 ns, Heating for 15 ns, Equilibration for 15 ns and Production for 30 ns.  Finally 25 snapshots of the trajectory were taken and superimposed into a single structure using the software VMD 1.3, as shown in Figure 2. RMSF was calculated for the residue index and depicted in Figure 3 and the best structure was taken by protein structure assessment tools such as procheck, ERRAT and Ramachandran plot and results were given in Table 2. (cf. Page 5)

Comment #6: This comment has not been addressed in the manuscript. It is still not clear to the reader why some compounds such as 4, 9, and 22 form two hydrogen bonds, yet their binding energy is less favourable than those of some compounds that form only one hydrogen bond. Additionally, some compounds do not form any hydrogen bond (i.e. compound 10) and they still have more favourable binding energy than some of the compounds that form one hydrogen bond. Authors should comment and explain those observations.

Response: The comment has been addressed in the revised manuscript (Table 4) Page-9 Ans: The docking score (binding energy) is not based on hydrogen bond interactions only. The binding energy is also based on the other supramolecular interactions and also the particular interactions in the active site of amino acids. Validating the H-bond interaction is not done in the present investigation so it is difficult to comment and/or conclude on the number of H-bond interactions and binding energies. The calculation is designed in a such a way that the upper bound and lower bound which optimized with lower energy will possess the best score.  Ref: Pantsar, T. and Poso, A., 2018. Binding affinity via docking: fact and fiction. Molecules, 23(8), p.1899.

Comment #7: Analysis of hydrophobic and pi-pi interactions is still missing. It can be performed on the already obtained complexes.

Response: Non bonded contacts such as pi-pi interactions, hydrophobic interactions and aromatic interaction were also estimated using PBD sum server and listed in the table 4. A greater number of interactions was observed for Caffeic-Acid – 12, (3R, 3aS, 6R, 6aR)-3-(2H-1, 3-benzodioxol-4-yl)-6-(2H-1,3-benzodioxol-5-yl)-hexahydrofuro[3,4-c]furan-1–11 and Rosmarinic Acid – 9. Similarly even though few compounds having better binding affinity do not show hydrogen bonds, they have seem to have some non-bonded interactions; namely, 2,4,7,9-Tetramethyl-5decyn4,7diol – 8, Diallyl tetrasulfide – 5 and 1,3-Cyclopentadiene – 3. (cf. Page 10)

Comment #12: Authors indeed added date, when the database was accessed (17/10/2021). However, this date is not correct one. The date when the database was accessed should be the one when it was accessed for the purpose of the study and not just when the manuscript was corrected.

Response: According to reviewer comments we removed the accessed date and mentioned reference site. https://phytochem.nal.usda.gov/phytochem/search (Page 2)

Comment #20: I believe the Figure 2 still does not tell the reader anything about the active site of the enzyme.

Response: As per the reviewer comment, Figure 2 was modified in the revised manuscript. The active site of enzyme is very important for docking and Fig. 2 depicts the grid box selection for the protein. Selection of the grid box in molecular docking is very important.

Comment #21: Binding energies and Ki are still written with different number of decimal numbers. Also hydrogen bond lengths in the main text are still written with too high precision.

Response: As pointed out by the reviewer, the changes were made in the revised manuscript (cf. Table 4, Page 9).

Comment #22: Figures 3-5 were not adapted in a clear enough way (interacting residues are still not named in the Figure and are shown only as ribbons, which does not explain the type of interaction to the reader).

Response: As suggested by the reviewer, the interacting residues were named and included in Figs 6-8. (Pages 11-12)

In summary, the authors wish to acknowledge Reviewer #1 for the insightful and constructive comments, along with the opportunity to improve the overall quality of this submission. The corresponding changes have been incorporated in the revised manuscript and we have further edited the submission for language, syntax, and clarity throughout to meet the high standards of this journal.

Reviewer 2 Report

The authors have improved the manuscript with their revisions and adequately addressed all points raised with one exception, listed below. 

The description in figure 1 is missing.

Checking the structure on the RCSB database, the enzyme consists of 6 chains (from A to F), but the authors only describe 3 chains (A, B and C) in the caption. Furthermore, in figure 1 there are two representations of the same protein without any explanation.

A figure in a manuscript has the function of illustrating a concept expressed in the text.

If the authors cannot improve on this, then figure 1 should be removed.

Author Response

Author Response to Reviewer comments on MS ID: micromachines-1387081

Reviewer #2 (round 2)

Comments and Suggestions for Authors

The authors have improved the manuscript with their revisions and adequately addressed all points raised with one exception, listed below. 

The description in figure 1 is missing.

Response: Figure 1 was removed from the manuscript.

Checking the structure on the RCSB database, the enzyme consists of 6 chains (from A to F), but the authors only describe 3 chains (A, B and C) in the caption. Furthermore, in figure 1 there are two representations of the same protein without any explanation.

Response: According to the PDB file 5A8K, the protein has six chains; namely, A, B, C, D, E and F. The protein chains; namely D, E and F were a repetition of Chains A, B and C, respectively. (cf. Page 3)

Figure in a manuscript has the function of illustrating a concept expressed in the text.

If the authors cannot improve on this, then figure 1 should be removed.

Response: As recommended by the reviewer, Fig 1 has been removed in the revised manuscript.

In summary, the authors wish to acknowledge Reviewer #2 for the insightful and constructive comments, along with the opportunity to improve the overall quality of this submission. The corresponding changes have been incorporated in the revised manuscript and we have further edited the submission for language, syntax, and clarity throughout to meet the high standards of this journal.

Round 3

Reviewer 1 Report

I appreciate the work done by Jothiramalingam et al., especially the MD simulations they performed now. I believe that performed simulations increased the quality of the results significantly.

The only comment I have is the Figure 6, it is of poor quality, I believe the number of dpi should be increased.